# Prevalence and correlates of symptoms of post-traumatic stress disorder among Chinese healthcare workers exposed to physical violence: a cross-sectional study

Lei Shi,[1] Lingling Wang,[2] Xiaoli Jia,[2] Zhe Li,[1] Huitong Mu,[1] Xin Liu,[1] Boshi Peng,[3] Anqi Li,[1] Lihua Fan[1]

[1]Department of Health Management, School of Public Health, Harbin Medical University, Harbin, Heilongjiang, China
[2]Department of Autonomous Protection, Chinese Hospital Association, Beijing, China
[3]Department of Scientific Research, The Second Affiliated Hospital of Harbin Medical University, Harbin, Heilongjiang, China

**Correspondence to**
Professor Lihua Fan;
lihuafan@126.com

## ABSTRACT

**Objectives** Post-traumatic stress disorder (PTSD) is a common psychological maladjustment to undergoing a traumatic event. Our aim was to measure the prevalence of PTSD among Chinese healthcare workers exposed to physical violence, and explore the associations of their demographic characteristics, social support, personality traits, and coping styles with their PTSD symptoms.

**Methods** A cross-sectional study was conducted using the Workplace Violence Scale, Posttraumatic Stress Disorder Checklist-Civilian Version (PCL-C), Social Support Rating Scale (SSRS), Eysenck Personality Questionnaire-Revised Short Scale and Trait Coping Style Questionnaire. We used convenience sampling method to collect data from March 2015 to September 2016. Healthcare workers (n=2706) from 39 public hospitals located in Heilongjiang, Hebei and Beijing provinces of China completed the questionnaires (effective response rate=84.25%).

**Results** Overall, the prevalence of physical violence in the previous 12 months was 13.60% (n=2706). The prevalence of PTSD among the healthcare workers who experienced physical violence was 28.0% (n=368). Most of the victims of physical violence (50.80%) did not exhibit PTSD symptoms based on their PCL-C scores, and 47.0% did not manifest the diagnostic criteria for PTSD after experiencing physical violence. The level of PTSD symptoms was negatively correlated with their scores on the SSRS (r=−0.188, p<0.001). The hierarchical regression analysis (block 3) revealed that, in women, positive coping was significantly associated with PTSD symptoms (β=−0.376, p=0.001). However, the effect of positive coping was not significant in men.

**Conclusions** The results suggest that the aftermath of physical violence contributes to the current prevalence of PTSD. The positive effects of social support on PTSD symptoms suggest that it has practical implications for interventions to promote psychological health. The healthcare workers' coping styles influenced the development of PTSD symptoms. Therefore, adopting effective coping styles and receiving social support have potential roles in the recovery from trauma after experiencing physical violence.

### Strengths and limitations of this study

► In China, few studies have been conducted on post-traumatic stress disorder (PTSD) symptoms following healthcare workers' exposure to physical violence.
► We assessed the prevalence of PTSD and explored the correlates of PTSD symptoms among Chinese healthcare workers exposed to physical violence.
► Our study was conducted at 39 public hospitals in three provinces using convenience sampling. Therefore, the representativeness of the sample is limited.
► The retrospective approach to collecting data using self-reports of PTSD symptoms might have led to recall and report bias.

## BACKGROUND

Post-traumatic stress disorder (PTSD) is a psychological state of imbalance, characterised by a series of chronic emotional reactions to a traumatic event, including re-experiencing, avoidance and heightened arousal, as outlined in the Diagnostic and Statistical Manual of Psychiatric Disorders fourth edition (DSM-IV).[1-3] However, the criteria for PTSD in the manual's fifth edition (DSM-5) include not three but four symptom clusters: re-experiencing, avoidance, negative alterations in mood and cognition, and hyper-arousal.[4] It is worth noting that PTSD has shifted from its classification as an anxiety disorder in DSM-IV to a new category of trauma and stress-related disorders in DSM-5.[4] Although a substantial number of studies indicate that almost all people exhibit intrusive and repetitious symptoms after exposure to excessive stress,[5] only a small percentage develop avoidance and hyper-arousal symptoms. Most individuals showing PTSD symptoms after exposure

to a traumatic event recover within weeks or months. However, 10%–25% might develop chronic PTSD that lasts for several months or years, or even a lifetime.[6]

PTSD originated from reports of the war-related trauma and was applied gradually to a variety of man-made and natural disasters.[7] Scholars have reported that the incidence of PTSD among male and female Vietnam veterans in the USA is 15.2% and 8.5%, respectively.[8] Moreover, most of the Chinese studies on PTSD have focused on wars, traffic accidents and natural disasters.[9 10] Differences in the incidence rates of PTSD for different types of trauma have been reported in China. For instance, the prevalence of PTSD has been reported to be 8.65% among soldiers assigned to military vehicles at high altitudes, 33.89% among flood-disaster survivors, 18.8% among earthquake survivors, 41% among traffic-accident survivors and 78.6% among survivors of serious explosions.[11]

PTSD symptoms and the full range of criteria comprising a PTSD diagnosis have been observed in rescue and ambulance personnel.[12 13] Healthcare workers typically are exposed to two types of trauma in the hospital setting: direct (personal involvement in traumatic events through confrontations resulting in their own traumatic experiences, e.g., workplace violence) and indirect (non-personal involvement in traumatic events through others' confrontations resulting in other people's traumatic experiences, e.g., witnessing other people's direct experiences of workplace violence, caring for dying patients, and threats of severe injury or exposure to trauma).[4 14–16] In the present study, a traumatic event refers to a healthcare worker's exposure to physical violence in the workplace. Workplace violence is divided into physical and psychological violence.[17] Physical violence causes more serious physical and psychological damage (e.g., PTSD, anxiety, fear and depression) than other forms of violence.[18–20] Physical violence refers to the use of physical force against an individual or a group, and can lead to physical, psychological or sexual harm; it includes hitting, shooting, kicking, slapping, pushing, biting, pinching, wounding using sharp objects, and sexual assault and rape.[17] Approximately 50% of healthcare workers have experienced at least one violent incident during their working lives.[21] During the past 12 months, the incidence rate of physical violence for nurses in different countries has ranged from 9.1% to 56.0%.[22–25] The results of a systematic review of studies conducted in Iran indicated that the most common types of physical violence experienced by 43% of participants were pushing or pinching.[26] In China, physician–patient conflicts present a growing trend, with an increase in the number of healthcare workers killed by patients or their relatives to 24, and an increase in injures from 2003 to 2013.[27] Several studies have estimated the prevalence of PTSD among emergency department staff to range from 10% to 25%.[28–30] Robertson and Perry conducted a systematic review of PTSD research investigations; the results showed that the prevalence of PTSD ranged from

8% to 29% among different hospital-based departments.[31] There are also reports of the occurrence of PTSD among Chinese nurses working in emergency departments, intensive care units, and operating rooms. However, the number of research studies on PTSD among healthcare workers has been relatively few in China.

Demographic variables (e.g., age, gender and educational level) and psychological and social variables (e.g., personality, coping style and social support) have been found to be significantly associated with cancer-related PTSD symptoms.[32 33] Previous studies have found that the risk of PTSD was most strongly associated with neuroticism and problem-focused coping strategies in the general population.[34 35] Neuroticism was the most critical personality dimension in predicting PTSD, and avoidant coping and social support mediated the relationship between neuroticism and PTSD in a high proportion of adult burn survivors.[36] Social support has been reported to play a significant role in helping nurses cope with work-related stress.[37] A meta-analysis indicated that work-related critical incidents were positively related to PTSD symptoms.[38]

In this study, we aimed to assess the prevalence of PTSD, and to explore the associations of demographic characteristics, social support, personality characteristics, and coping styles with PTSD symptoms among Chinese healthcare workers exposed to physical violence.

## METHODS
### Participants and procedures
A cross-sectional study was conducted from March 2015 to September 2016 with a sample of healthcare workers employed by 39 public hospitals located in Heilongjiang, Hebei, and Beijing provinces of China. The 39 public hospitals that served as the research settings were chosen using convenience sampling. All investigators were trained using a uniform survey manual before they began to collect data. Qualified investigators were appointed to collect data. We obtained permission from the managers and the medical dispute resolution and human resources departments of the hospitals. The investigators conducted face-to-face surveys by using an anonymous, self-administered questionnaire. We purposely selected three public hospitals in Harbin (the First Affiliated Hospital of Harbin Medical University, the Second Affiliated Hospital of Harbin Medical University and the Fourth Affiliated Hospital of Harbin Medical University) as the sites for our pilot study before the formal investigation. A total of 150 questionnaires were distributed and returned (these data were excluded from the main study). A total of 3212 healthcare workers (i.e., physicians, nurses and medical technicians) were investigated using convenience sampling in the formal investigation. The researchers and hospital coordinators distributed and collected the questionnaires that were completed by the healthcare workers immediately. A total of 2706 valid questionnaires were returned, and the effective response rate was 84.25%. This study's focus was only on PTSD

symptoms among healthcare workers exposed to physical violence; thus, only 368 responses were considered valid data and were analysed in the present study.

The inclusion criteria for participation in this study were as follows: (1) at least 1 year of work experience, (2) voluntary participation, (3) participation would not affect the participant's work and (4) experience of physical violence in the previous 12 months. Individuals were excluded if they (1) had received any psychological treatment after experiencing physical violence; (2) experienced other traumatic events, including workplace psychological violence or serious life events (e.g., domestic violence or attacks by criminals), serious accidents (e.g., fires, explosions or traffic accidents), natural disasters (e.g., typhoons, earthquakes or floods); or (3) were indirectly exposed to trauma[15][39] (e.g., witnessing other people experience traumatic events).

## Questionnaire

### Demographic characteristics

Demographic data on the healthcare workers were collected, including gender, age, marital status, educational status, professional title, department, occupation and work experience. Age was categorised as ≤30, 31–50 and ≥51 years old. Marital status was categorised as married and single/divorced/widowed. Educational status was classified as junior college or below, undergraduate and graduate. Occupation was divided into three groups: physician, nurse and medical technician. Professional title was categorised as primary, intermediate and senior. Department was classified as emergency department, internal medicine, surgery, obstetrics and gynaecology, paediatrics and other. Work experience was divided into four categories: ≤4, 5–10, 11–20, and ≥21 years.

### Workplace Violence scale

The Workplace Violence (WPV) Scale developed by the International Labour Office, International Council of Nurses, WHO and Public Services International Joint Programme on Workplace Violence in the Health Sector in 2003 and the revised Survey of Violence Experienced by Staff were used to measure workplace violence.[40][41] We obtained permission to use these scales. The scale used in this study consists of two dimensions (physical violence and psychological violence) and has nine items that were adopted from these scales. Each item is scored on a 4-point scale reflecting respondents' frequency of exposure to violence (0=0 times, 1=1 time, 2=2–3 times and 3=≥4 times). The total possible score ranges from 0 to 27, with a higher total score indicating a higher frequency of exposure to WPV. The physical violence subscale consists of six items; thus, the total possible score ranges from 1 to 18. In the present study, Cronbach's α for the WPV Scale was 0.86.

### Post-traumatic stress disorder

The PTSD Checklist-Civilian Version (PCL-C), which has been used to measure PTSD symptoms among healthcare workers, was used in the present study.[42][43] It consists of 17 self-report items, which comprise three dimensions, namely, re-experiencing, avoidance/numbing and hyper-arousal. The three dimensions correspond to the DSM-IV symptoms criteria for PTSD.[2] The response options for each item on PCL-C are rated from 1 (not at all) to 5 (extremely), based on the extent to which the respondent has been troubled by specific symptoms in the past month. The total possible score is calculated by adding the scores for all items, and it ranges from 17 to 85 points, with a higher score indicating a higher risk for PTSD symptoms. A total score≥50 is indicative of the full PTSD diagnosis (sensitivity=0.82; specificity=0.83; kappa=0.64).[44] In this study, the traumatic event in the original PCL-C was replaced by physical violence. The reliability and validity of this instrument have been shown to be high in a wide range of Chinese samples.[45] The present study revealed that Cronbach's α for PCL-C was 0.934, and for the three subscales it was 0.872 (re-experiencing), 0.921 (avoidance) and 0.926 (hyper-arousal).

### Eysenck Personality Questionnaire-Revised Short Scale

Personality traits were measured using the Eysenck Personality Questionnaire-Revised Short Scale for Chinese (EPQ-RSC).[46][47] EPQ-RSC consists of 48 items, categorised into four subscales reflecting personality traits: extraversion, neuroticism, psychoticism and lie. Each item is scored on a dichotomous scale (1=Yes, 0=No) to measure personality characteristics. The scores of the positively and negatively worded items are summed in accordance with each personality trait. Early studies found EPQ-RSC to have high reliability and validity as a measure of personality traits in China.[47][48] The total score for the extraversion subscale indicates introversion when it is less than 43.3, intermediate when it is from 43.3 to 56.7, and extraversion when it is greater than 56.7.[48] For the psychoticism subscale, tough-minded is defined as a total score greater than 56.7, intermediate is defined as a total score between 43.3 and 56.7, and mild is defined as a total score less than 43.3.[48] For the neuroticism subscale, a total score of less than 43.3 defines emotional stability, whereas a total score from 43.3 to 56.7 defines intermediate, and a total score greater than 56.7 defines emotional instability.[48] For the lie subscale, a total score of 60 or greater indicates that information provided by the respondent might be unreliable.[48] In this study, Cronbach's α for EPQ-RSC was 0.903. The internal consistency coefficients were 0.854, 0.756, 0.791 and 0.762, for the extraversion, neuroticism, psychoticism and lie subscales, respectively.

### Trait Coping Style Questionnaire

The Trait Coping Style Questionnaire (TCSQ) was used in this study to assess participants' style of coping with life events.[37][49] TCSQ consists of 20 items, including 10 items measuring positive coping and 10 items measuring negative coping. Positive coping refers to individuals who, when faced with a problem, tend to deal with it in a positive way and are able to quickly forget unpleasant aspects. Negative coping refers to the tendency to use negative coping methods to deal with problems and vent

frustrations to other people, which makes it is easier to ignore unpleasant thoughts. For example, when conflicts with others arise, individuals who use negative coping will ignore the opposing side for a long time.[50] Each item is rated on a 5-point Likert Scale. The total possible score for positive and negative coping is calculated by adding the scores for all the items. Previous studies have found TCSQ to have high reliability and validity as a measure of coping style in China.[49 50] In this study, Cronbach's α for the total scale was 0.845, and the internal consistency coefficients of the subscales were α=0.823 (positive coping) and α=0.863 (negative coping).

### Social Support Rating Scale

Social support was evaluated using the Chinese version of the Social Support Rating Scale (SSRS),[51–53] which is a brief measure of the overall situation of respondents' social support. This 10-item scale is divided into three dimensions: subjective support, objective support and utilisation of support. Subjective support refers to an individual's emotional experience of being respected, supported and understood by their social group, and it is closely related to the individual's subjective feelings. Objective support refers to visible support, including material and direct assistance, social networks, group relationships and the individual's degree of participation in societal activities with family, friends and colleagues (eg, marriage). A low level of social support is defined as a total score between 12 and 44, an intermediate level as a total score between 45 and 54 and high level as a total score greater than 55.[53] The present study revealed that Cronbach's α for SSRS was 0.865, and for the three subscales, it was 0.884 (subjective support), 0.911 (objective support) and 0.875 (availability of support).

### Data analysis

EpiData V.3.1 was used to establish the study's database. We eliminated the questions with missing data or quality issues. To ensure accuracy, two trained personnel entered the data after all the surveys were completed. IBM SPSS version 19.0 and Excel were used for the data analysis. The normal distributions of the continuous variables were verified using P–P plots and K-S tests. Descriptive statistics, including numbers (n), percentages (%), means and SD were calculated for the demographic variables. We used one-way analysis of variance or independent sample t-tests to compare group differences on the measures of the continuous variables. The $\chi^2$ test was used to compare differences in the categorical variables. Pearson's correlations were used to examine correlations among the continuous variables. Hierarchical regression analysis was used to examine the associations of the demographic characteristics and the scores on SSRS, EPQ-RSC and TCSQ with PTSD symptoms. Statistics, including F values, $R^2$ changes ($\Delta R^2$),[22] standardised regression coefficients (β) and p values, for each step in the regression model were reported. All the study variables were tested for multicolinearity. A p value <0.05 was considered to be statistically significance.

### Strengthening the Reporting of Observational Studies in Epidemiology statement

We declared that the Strengthening the Reporting of Observational Studies in Epidemiology guidelines are followed in this study.

### Ethical considerations

Ethical approval to conduct this study was granted by the research ethics committee of Harbin Medical University, and informed consent was obtained from each hospital and healthcare worker involved in the investigation. All of the participants gave their informed consent before the survey; they were assured that their personal information would be kept confidential.

## RESULTS

### Demographic characteristics of the respondents

The demographic characteristics of the respondents are shown in table 1.

### Characteristics of the victims in relation to PTSD symptoms

Of the 368 victims of physical violence, 59.8% were women, 51.3% completed an undergraduate education and 73.9% were married. The characteristics of the victims in relation to PTSD symptoms are presented in table 2.

### Prevalence of physical violence in the previous 12 months

During the past 12 months, the prevalence of physical violence and psychological violence toward healthcare workers was 13.60% (368/2706) and 59.64% (1614/2706), respectively. The respondents reported that the patients' relatives were the main perpetrators (67.4%, n=248), followed by the patients (23.6%, n=87).

### Prevalence of PTSD

The PTSD symptoms based on the victims' PCL-C scores are summarised in table 3. According to their scores on PCL-C, 103 victims (28.0%) met the full criteria for a PTSD diagnosis and 21.2% of victims were at risk for developing PTSD.

According to the DSM-IV text revision (TR) criteria for PTSD,[2] 47.0% of the victims did not appear to manifest the diagnostic criteria. Re-experiencing was the most frequently observed criterion for PTSD observed among the victims (45.1%), followed by hyper-arousal (37.8%).

### Correlations of the EPQ-RSC, TCSQ and SSRS scores with PTSD symptoms

Table 4 shows the correlations among the victims' PTSD symptoms and scores on EPQ-RSC, TCSQ, SRSS and the physical violence subscale. The mean score for PTSD symptoms on PCL-C was 40.60 (SD=15.01). As expected, the level of PTSD symptoms was negatively correlated with respondents' scores on SSRS (r=−0.188, p<0.001) and positive coping subscale of TCSQ (r=−0.164, p=0.002), respectively. Physical violence was positively associated with PTSD symptoms (r=0.259, p<0.001). The level of PTSD symptoms was positively correlated with victims'

**Table 1** Demographic characteristics of the respondents (n=2706)

| Demographic variables | | n | % |
| --- | --- | --- | --- |
| Gender | | | |
| | Male | 623 | 23.0 |
| | Female | 2083 | 77.0 |
| Age group | | | |
| | ≤30 | 1258 | 46.5 |
| | 31–50 | 1341 | 49.5 |
| | ≥51 | 107 | 4.0 |
| Educational level | Junior college or below | 856 | 31.6 |
| | Undergraduate | 1341 | 49.6 |
| | Graduate | 509 | 18.8 |
| Marital status | Married | 1859 | 68.7 |
| | Single/divorced/widowed | 847 | 31.3 |
| Occupation | Physician | 1058 | 39.1 |
| | Nurse | 1520 | 56.2 |
| | Medical technician | 128 | 4.7 |
| Technical title | Primary | 1147 | 42.4 |
| | Intermediate | 1026 | 37.9 |
| | Senior | 533 | 19.7 |
| Department | Emergency department | 323 | 11.9 |
| | Internal medicine | 813 | 30.0 |
| | Surgery | 752 | 27.8 |
| | Obstetrics and gynaecology | 276 | 10.2 |
| | Paediatrics | 218 | 8.1 |
| | Other | 324 | 12.0 |
| Years of experience | ≤4 | 1014 | 37.5 |
| | 5–10 | 820 | 30.3 |
| | 11–20 | 503 | 18.6 |
| | ≥21 | 369 | 13.6 |

scores on the negative coping subscale of TCSQ (r=0.188, p<0.001).

### Hierarchical regression analysis of factors related to PTSD symptoms

The results of the hierarchical regression analysis are presented in table 5. Variables that had a statistically significant association with PTSD were used as control variables. Gender had a significant effect on PTSD symptoms in the model (block 1). As shown in block 2, physical violence was positively associated with PTSD symptoms (β=1.216, p<0.001). As shown in block 3, positive coping as measured by the TCSQ was negatively associated with PTSD symptoms (β=−0.327, p=0.002), whereas, negative coping was positively associated with PTSD symptoms in the regression model (β=0.353, p=0.001). Furthermore, gender had a significant effect on PTSD symptoms, and men were more vulnerable to PTSD symptoms than

women (table 1). Therefore, we explored the potential correlates of PTSD symptoms in men and women (table 6). As shown in block 3, among the women, positive coping as measured by TCSQ, was significantly associated with PTSD symptoms (β=−0.376, p=0.001), but the effect of positive coping was not significant in men.

### DISCUSSION

In this cross-sectional hospital-based study of healthcare workers exposed to physical violence, we assessed the prevalence and correlates of PTSD symptoms. Our study found that the prevalence of physical violence among the healthcare workers was approximately 13.6% in the previous year. The results of a study conducted during 2009–2010 in Italy found that 13.4% of nurses were exposed to physical violence,[22] which is similar to the frequency found in this study. However, other studies have reported higher prevalence rates of physical violence than the current study.[23 24] The inconsistency in these findings may be attributed to cultural differences between countries or missing reports. PTSD was reported by 28.0% of the victims based on the scoring instructions of PCL-C (i.e., 28.0% scored 50 points and above). We selected the PCL-C score of 50 and above as the standard cut-off due to the influence of traditional Chinese culture on the frequency of healthcare workers' encounters with traumatic events and the DSM-IV-TR criteria for PTSD.[2] Previous studies have provided valuable information regarding the prevalence of PTSD among doctors and nurses.[28–31] The prevalence of PTSD among the healthcare workers exposed physical violence in our study was similar to that reported in Atlanta.[54] However, the prevalence rates of PTSD in these studies were different from the present study,[55 56] which might be attributed to differences in the studies' sample characteristics, designs, definitions and diagnostic criteria for PTSD, due to their varied cultural backgrounds. Moreover, the prevalence of PTSD symptoms in our sample was higher than that of the general population (8%) in the USA.[57] This finding might be attributed to the fact that the general population's frequency of exposure to serious traumatic events is lower than that of healthcare workers. Similarly, nurses who work in intensive care units experience traumatic events more often than other healthcare workers do.[29]

Our study found that 21.2% of the victims of physical violence were at risk for developing PTSD and 28.0% met the full diagnostic criteria for PTSD. This finding suggests that physical violence had a strong influence on the mental health of healthcare workers. Approximately 53.0% (195/368) of the victims reported having at least one PTSD criterion. The most commonly observed PTSD symptom was re-experiencing (45.1%), followed by hyper-arousal (37.8%), and then avoidance (35.1%). A previous study also reported that healthcare workers in an emergency department were victims of direct workplace violence because they reported re-experiencing the violent event, followed by hyper-arousal and avoidance.[28]

**Table 2** Characteristics of victims in relation to PTSD symptoms (n=368)

| | Variables | n | % | PTSD symptoms | | F/t | p |
|---|---|---|---|---|---|---|---|
| | | | | Mean | SD | | |
| Gender | Male | 148 | 40.2 | 44.03 | 16.19 | 3.537 | 0.000 |
| | Female | 220 | 59.8 | 38.30 | 13.71 | | |
| Age group | ≤30 | 133 | 36.1 | 38.09 | 13.36 | 2.946 | 0.054 |
| | 31–50 | 216 | 58.7 | 42.01 | 15.56 | | |
| | ≥51 | 19 | 5.2 | 42.10 | 17.80 | | |
| Educational level | Junior college or below | 118 | 32.1 | 38.14 | 13.54 | 2.592 | 0.076 |
| | Undergraduate | 189 | 51.3 | 42.13 | 15.46 | | |
| | Graduate | 61 | 16.6 | 40.64 | 15.85 | | |
| Marital status | Married | 272 | 73.9 | 41.51 | 15.75 | 2.195 | 0.029 |
| | Single/divorced/widowed | 96 | 26.1 | 38.03 | 12.38 | | |
| Occupation | Physician | 175 | 47.6 | 42.97 | 15.37 | 4.379 | 0.013 |
| | Nurse | 180 | 48.9 | 38.29 | 13.82 | | |
| | Medical technician | 13 | 3.5 | 40.69 | 21.24 | | |
| Technical title | Primary | 145 | 39.4 | 39.56 | 13.04 | 0.576 | 0.562 |
| | Intermediate | 126 | 34.2 | 41.32 | 16.21 | | |
| | Senior | 97 | 26.4 | 41.23 | 16.16 | | |
| Department | Emergency department | 68 | 18.5 | 41.46 | 16.08 | 0.722 | 0.607 |
| | Internal medicine | 76 | 20.7 | 38.45 | 15.07 | | |
| | Surgery | 123 | 33.4 | 41.53 | 13.91 | | |
| | Obstetrics and gynaecology | 19 | 5.2 | 41.63 | 16.52 | | |
| | Paediatrics | 27 | 7.3 | 37.63 | 10.37 | | |
| | Other | 55 | 14.9 | 41.55 | 17.28 | | |
| Years of experience | ≤4 | 101 | 27.4 | 37.19 | 13.25 | 2.158 | 0.063 |
| | 5–10 | 120 | 32.6 | 42.13 | 14.52 | | |
| | 11–20 | 87 | 23.7 | 41.90 | 16.80 | | |
| | ≥21 | 60 | 16.3 | 41.42 | 15.46 | | |
| Social support | Low | 224 | 60.9 | 42.41 | 15.06 | 5.904 | 0.003 |
| | Medium | 130 | 35.3 | 38.52 | 14.40 | | |
| | High | 14 | 3.8 | 31.00 | 14.53 | | |
| Subjective support | Low | 22 | 6.0 | 41.91 | 18.63 | 0.859 | 0.425 |
| | Medium | 38 | 10.3 | 43.37 | 12.93 | | |
| | High | 308 | 83.7 | 40.17 | 14.97 | | |
| Objective support | Low | 206 | 56.0 | 42.19 | 15.47 | 3.369 | 0.035 |
| | Medium | 155 | 42.1 | 38.87 | 14.32 | | |
| | High | 7 | 1.9 | 32.00 | 9.24 | | |
| Utilisation of support | Low | 39 | 10.6 | 48.18 | 16.98 | 6.979 | 0.001 |
| | Medium | 259 | 70.4 | 40.37 | 14.56 | | |
| | High | 70 | 19.0 | 37.24 | 14.23 | | |
| Extraversion | Introversion | 102 | 27.7 | 39.45 | 13.35 | 1.278 | 0.280 |
| | Middle | 164 | 44.6 | 41.99 | 16.04 | | |
| | Extraversion | 102 | 27.7 | 39.51 | 14.80 | | |
| Psychoticism | Mild | 68 | 18.5 | 42.22 | 16.87 | 0.998 | 0.370 |
| | Middle | 213 | 57.9 | 40.79 | 15.24 | | |
| | Tough-minded | 87 | 23.6 | 38.86 | 12.69 | | |

Continued

**Table 2** Continued

| | Variables | n | % | PTSD symptoms | | F/t | p |
| --- | --- | --- | --- | --- | --- | --- | --- |
| | | | | Mean | SD | | |
| Neuroticism | Emotional instability | 100 | 27.2 | 40.33 | 13.80 | 0.530 | 0.589 |
| | Middle | 153 | 41.6 | 41.50 | 16.72 | | |
| | Emotional stability | 115 | 31.2 | 39.63 | 13.60 | | |

PTSD, post-traumatic stress disorder.

Laposa and Alden reported that re-experiencing an incident of physical violence was significantly and negatively associated with emergency department workers' ability to accomplish their work.[28] It is possible that the prevalence of symptoms of hyper-arousal and avoidance was not higher due to the distinctive characteristics of the healthcare workers' jobs and the hospital's culture, which required them to be able to shift their focus quickly and constantly. Healthcare workers who escaped slight injury during an episode of physical violence had to shift their focus rapidly to another patient after the event, and they could not avoid the work environment.[28]

As shown in the results of the Pearson's correlations and the hierarchical regression analysis, social support had a significant negative association with PTSD symptoms, and this finding is consistent with other researches.[9 36 51 52] The level of PTSD symptoms was significantly and negatively correlated with the healthcare workers' scores for objective support and utilisation of support. A previous study found that the Deterioration Model of Social Support has been useful in discriminating the potential of stressors to reduce support.[57] They found that disaster-induced erosion of perceived social support increased symptoms of depression among both primary and secondary victims; the loss of perceived social support also mediated psychological consequences.[58] The Deterioration Deterrence

**Table 3** Sample description and prevalence of PTSD

| PTSD symptoms | Physical violence | |
| --- | --- | --- |
| | n | % |
| PTSD symptoms based on PCL-C scores | | |
| No obvious PTSD symptoms (17–37) | 187 | 50.8 |
| Criteria met for potential risk of PTSD (38–49) | 78 | 21.2 |
| Criteria met for the full PTSD diagnosis (50–85) | 103 | 28.0 |
| PTSD symptoms based on PTSD criterion* | | |
| No criterion manifestation | 173 | 47.0 |
| Re-experiencing (criterion B) | 166 | 45.1 |
| Avoidance (criterion C) | 129 | 35.1 |
| Hyper-arousal (criterion D) | 139 | 37.8 |

*Participants may have more than one criteria.
PCL-C, Posttraumatic Stress Disorder Checklist-Civilian Version; PTSD, post-traumatic stress disorder.

Model of Social Support, which is similar to support-mobilisation models, has been used to explain how the perceived deterioration of social support can be counteracted by higher levels of received social support.[58 59] If post–disaster support mobilisation is implemented, stress should be positively correlated with received support. At the same time, received support should be positively related to perceived support. Therefore, the receipt of support should suppress a negative relationship between stress and perceived support.[58 59] Victims of physical violence should be encouraged not to abandon their daily social activities because these activities have many important functions (e.g., they help people understand the needs of network members and inspire their participation in helping).[59] Daily contact is the most natural forum for sharing experiences, which might suppress negative emotions, provide opportunities for social comparison, and maintain a sense of friendship and feelings of being accepted.[59] It is important to recognise that stress caused by violence is persistent. Yet, a supportive hospital environment can help individuals cope with a wide range of stressful events and serve as a buffer against their negative health effects.[58 59]

Another significant effect of coping styles on PTSD symptoms was found in the present study. This result indicated that when healthcare workers encountered a traumatic event, a negative coping style was more likely to increase their proneness to developing PTSD symptoms. This finding is consistent with the results of other studies.[36 49 52] Positive coping was beneficial in preventing or alleviating PTSD symptoms in our study. A previous investigation found that active coping had also a positive relationship with PTSD.[36] Unexpectedly, the three personality factors were not significantly associated with PTSD symptoms. However, Lawrence and Fauerbach's study found that individuals with higher neuroticism scores exhibited more PTSD symptoms.[36]

An important finding of the present study was revealed in the univariate analyses. We found that the men exposed to traumatic events were more likely to exhibit PTSD symptoms than the women were. This result was different from the findings reported in earlier studies that women are more likely to develop PTSD symptoms.[6 17 19] This finding might be attributed to gender differences in responses to different traumatic events and in social networks.[60 61] This phenomenon also might be attributable to the fact that the injuries sustained by

**Table 4** Pearson correlations among PTSD symptoms, EPQ-RSC, TCSQ, SSRS and physical violence

| Variables | Mean | SD | 1 | 2 | 3 | 4 | 5 | 6 | 7 | 8 | 9 | 10 | 11 | 12 | 13 | 14 |
|---|---|---|---|---|---|---|---|---|---|---|---|---|---|---|---|---|
| PTSD symptoms | 40.60 | 15.01 | – | | | | | | | | | | | | | |
| Re-experiencing | 12.43 | 4.92 | 0.89** | – | | | | | | | | | | | | |
| Avoidance | 15.70 | 5.99 | 0.94** | 0.77** | – | | | | | | | | | | | |
| Hyper-arousal | 12.46 | 5.48 | 0.91** | 0.70** | 0.79** | – | | | | | | | | | | |
| Physical violence | 3.08 | 2.99 | 0.26** | 0.22** | 0.31** | 0.18** | – | | | | | | | | | |
| SSRS | 41.73 | 8.44 | –0.19** | –0.12** | –0.22** | –0.17** | –0.12** | – | | | | | | | | |
| Subjective support | 25.23 | 5.14 | –0.09 | –0.02 | –0.13* | –0.09 | 0.01 | 0.89** | – | | | | | | | |
| Objective support | 8.63 | 3.30 | –0.21** | –0.17** | –0.23* | –0.17** | –0.20** | 0.77** | 0.46** | – | | | | | | |
| Utilisation of support | 7.87 | 2.01 | –0.21** | –0.15** | –0.22* | –0.20** | –0.21** | 0.66** | 0.43** | 0.40** | – | | | | | |
| Positive coping of TCSQ | 30.05 | 7.22 | –0.16** | –0.15** | –0.12* | –0.18** | –0.13* | 0.10 | 0.04 | 0.10* | 0.15** | – | | | | |
| Negative coping of TCSQ | 26.92 | 7.33 | 0.19** | 0.10 | 0.19* | 0.22** | 0.04 | –0.31** | –0.26** | –0.29** | –0.17** | 0.12* | – | | | |
| Extraversion | 49.81 | 10.33 | –0.01 | –0.03 | 0.01 | –0.01 | 0.06 | –0.05 | –0.02 | –0.08 | –0.02 | –0.04 | 0.02 | – | | |
| Psychoticism | 50.10 | 9.81 | 0.06 | 0.06 | 0.06 | 0.04 | 0.00 | 0.02 | 0.02 | 0.02 | 0.06 | 0.02 | 0.04 | –0.02 | – | |
| Neuroticism | 50.07 | 10.34 | 0.03 | 0.04 | 0.01 | 0.05 | 0.03 | 0.03 | 0.01 | 0.05 | 0.04 | 0.01 | 0.04 | –0.09 | 0.20** | – |

*p<0.05, **p<0.01.
EPQ-RSC, Eysenck Personality Questionnaire-Revised Short Scale for Chinese; PTSD, post-traumatic stress disorder; SSRS, Social Support Rating Scale; TCSQ, Trait Coping Style Questionnaire.

**Table 5** Hierarchical regression for exploring the correlates of PTSD symptoms

| Variables | Block 1 (β) | Block 2 (β) | Block 3 (β) | Block 4 (β) |
|---|---|---|---|---|
| Gender | −4.663* | −3.282 | −3.060 | −3.012 |
| Marital status | −2.021 | −1.859 | −2.626 | −2.798 |
| Occupation | −1.274 | −1.918 | −2.494 | −2.414 |
| Physical violence | | 1.216** | 1.015** | 1.028** |
| SSRS | | | −0.193* | −0.192* |
| Positive coping of TCSQ | | | −0.327** | −0.325** |
| Negative coping of TCSQ | | | 0.353** | 0.361** |
| Extraversion | | | | −0.049 |
| Psychoticism | | | | 0.081 |
| Neuroticism | | | | 0.042 |
| F | 5.189** | 9.886** | 10.544** | 11.584** |
| $R^2$ | 0.041 | 0.098 | 0.170 | 0.246 |
| $\Delta R^2$ | 0.041 | 0.057** | 0.072** | 0.076* |

PTSD, post-traumatic stress disorder; SSRS, Social Support Rating Scale; TCSQ, Trait Coping Style Questionnaire.
*p<0.05, **p<0.01.

the men after experiencing physical violence were more severe than those of the women. After experiencing physical violence, the women were likely to receive more social support than the men, suggesting that women were more often regarded as a vulnerable group.

**Table 6** Hierarchical regression for exploring the correlates of PTSD symptoms in men and women, respectively

| | Variables | Mean (SD) | Block 1 (β) | Block 2 (β) | Block 3 (β) |
|---|---|---|---|---|---|
| Male | | | | | |
| n=148 | Physical violence | 3.60 (3.16) | 1.216** | 1.033* | 1.073** |
| | SSRS | 41.25 (9.32) | | −0.255 | −0.257 |
| | Positive coping of TCSQ | 30.75 (7.28) | | −0.298 | −0.318 |
| | Negative coping of TCSQ | 27.21 (6.61) | | 0.467* | 0.479* |
| | Extraversion | 49.18 (10.04) | | | 0.062 |
| | Psychoticism | 50.02 (8.62) | | | −0.282 |
| | Neuroticism | 49.99 (10.59) | | | −0.072 |
| | F | | 8.727** | 5.961** | 4.182** |
| | $R^2$ | | 0.056 | 0.143 | 0.173 |
| | $\Delta R^2$ | | 0.056** | 0.087** | 0.030 |
| Female | | | | | |
| n=220 | Physical violence | 2.73 (2.83) | 1.169** | 0.955** | 0.953** |
| | SSRS | 42.05 (7.79) | | −0.057 | −0.066 |
| | Positive coping of TCSQ | 29.58 (7.16) | | −0.376** | −0.376** |
| | Negative coping of TCSQ | 26.72 (7.78) | | 0.296* | 0.297* |
| | Extraversion | 50.03 (10.54) | | | −0.121 |
| | Psychoticism | 50.14 (10.56) | | | 0.006 |
| | Neuroticism | 50.12 (10.20) | | | −0.005 |
| | F | | 13.441** | 7.488** | 4.557** |
| | $R^2$ | | 0.058 | 0.122 | 0.131 |
| | $\Delta R^2$ | | 0.058** | 0.064** | 0.009 |

*p<0.05, **p<0.01.
PTSD, post-traumatic stress disorder; SSRS, Social Support Rating Scale; TCSQ, Trait Coping Style Questionnaire.

These findings suggest that social support, exposure to physical violence and coping styles are closely related to PTSD symptoms. Therefore, we recommend interventions based on the social cognitive theory.[62] For example, social support has been found to be an important protective factor in reducing stress and depression and improving health.[62] After the occurrence of a traumatic event, enabling function of social support can enhance self-efficacy, thereby promoting recovery from the trauma.[62]

The present study has several limitations. First, we used PCL-C to assess PTSD symptoms rather than a standard clinical diagnostic method. Consequently, the prevalence of PTSD might be overestimated. Second, the study's findings need to be replicated in a longitudinal study. Finally, convenience sampling is a non-probability sampling method and the results of this study are specific to Chinese healthcare workers exposed to physical violence in the past 12 months. Thus, the low representativeness of the sample due to the use of convenience sampling limits the generalisability of the results. The inclusion of healthcare workers from a wider range of careers in a more representative sample should contribute to the ability to generalise the results of future studies.

## CONCLUSIONS

The results suggest that the aftermath of physical violence contributes to the current prevalence of PTSD. The positive effects of social support on PTSD symptoms suggest that social support has practical implications for psychological interventions to promote mental health. Furthermore, the coping styles of the healthcare workers in this study influenced the development of PTSD symptoms. Therefore, it is imperative to use positive coping methods and to receive social support after experiencing a traumatic event.

**Acknowledgements** The authors thank all the healthcare workers, managers and Chinphenomenon also might bephenomenon also might beese Hospital Association for their assistaphenomenon also might bephenomenon also might bence and support for this project.

**Contributors** LS and LF designed the study. LS, LW, XJ, BP and LF collected data. ZL, LW, XJ, HM, XL and AL analysed the data. LS and LF drafted the manuscript. LS, ZL and LF revised the manuscript.

**Funding** This study was funded by the National Natural Science Foundation of China, grant number 71473063.

**Competing interests** None declared.

**Patient consent** Obtained.

**Provenance and peer review** Not commissioned; externally peer reviewed.

**Data sharing statement** No additional data are available.

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
