## [Reviewer comments · BMJ Open]

ARTICLE DETAILS

TITLE (PROVISIONAL)	Prevalence and correlates of symptoms of post-traumatic stress disorder among Chinese healthcare workers exposed to physical violence: a cross-sectional study
AUTHORS	Shi, Lei; Wang, Lingling; Jia, Xiaoli; Li, Zhe; Mu, Huitong; Liu, Xin; Peng, Boshi; Li, Anqi; Fan, Lihua

VERSION 1 - REVIEW

REVIEWER	Nicola Ramaciati Azienda Ospedaliera di Perugia, Perugia (Italy)
REVIEW RETURNED	20-Mar-2017

GENERAL COMMENTS	Thanks so much for the opportunity to review this article. The paper is very well written, very interesting and enjoyable to read. I suggest that in the list of references: the journal abbreviation for all references. Not just some.
--

REVIEWER	Magdalena Lesnierowska SWPS University of Social Sciences and Humanities, Department of Psychology
REVIEW RETURNED	10-Apr-2017

GENERAL COMMENTS	Prevalence and correlates of symptoms of post-traumatic stress disorder among Chinese healthcare workers exposed to physical violence : a cross-sectional study. This interesting manuscript describes a cross-sectional study on the prevalence and potential correlates of physical violence-related post-traumatic stress symptoms among Chinese healthcare workers – the occupational group considered to be at high risk for job-related traumatization. Specifically, the study investigates the predictive role of demographic characteristics, social support, personality traits, and coping styles for PTSD and its components (in the terms of posttraumatic stress disorder symptoms relevant to DSM-IV criteria for PTSD). The study is based on self-report data. The strengths of the study are: 1. The choice of participants – the healthcare staff may be particularly affected by job-related stress and its consequences such as PTSD, especially in the context of workplace exposure to physical violence. Authors clearly argue their choice and indicate the underrepresentation of research on this problem in both general and
---

Chinese specifically occupational health studies field.

2. The problem of psychological costs of working in human services, specifically when the work environment may be demanding and primarily traumatizing e.g. because of the risk of physical violence. Thus, it is still important to explore prevalence and contributors of job-related traumatization among healthcare and support staff. The results of this study indicate that gender, social support, coping style, and personality traits was significantly associated to PTSD symptoms. However, as the study is cross-sectional, longitudinal research should be conducted to establish the causal relationships of tested variables. Also, more advanced statistical models could be applied (e.g. moderation/mediation analysis) to understand not only which potential predictors are related to PTSD symptoms, but to explore when and how they contribute to PTSD development.

Beside the strengths, there are some concerns, not clearly outlined comprehensive conceptual framework and variables operationalization, surface literature review and results discussion, and simplified statistical model. Detailed concerns are listed below. Hopefully the authors will find these comments helpful.

1. The PTSD is defined (page 4) in the terms of DSM-IV-TR (APA, 2000). Please note, that the most current DSM edition is DSM-5 (APA, 2013), which should be also discussed in the context of trauma symptoms described in this manuscript. DSM-5 criteria for PTSD include not three but four symptom clusters. Also while defining, I might be worth stating that PTSD shifted from anxiety disorder in DSM-IV to new category: "Trauma and Stress-related Disorders" in DSM-5.

2. The theoretical framing for PTSD is superficially described. It is recommended to expand the conceptual part in terms of why and how posttraumatic stress symptoms develop, especially among this particular occupational group. The clear conceptual/theoretical framework would serve as a clue for understanding why do Authors choose a particular set of variables (demographics, social support, coping styles, personality traits) to be considered as PTSD correlates/predictors. It isn't clear why exactly these variables are tested while others are not. Presumably Authors implemented an eclectic approach, however it should be clearly outlined. If the choice of variables is evidence based, proper empirical background i.e. the review of study results should be provided instead of statement, that there are some studies that found the associations (what kind of associations? in what study populations?) between chosen variables and PTSD (p. 5). Provision of meta-analysis data would be highly appreciated.

3. Authors do not provide a reference and data on prevalence of PTSD symptoms among healthcare workers (p. 5), however refer to data on e.g. Vietnam veterans in USA (p. 4). It would be beneficial to refer to relevant results (preferably meta-analytical).

4. The study aimed to explore the prevalence of PTSD among healthcare works exposed to physical violence, however no operationalization of work-related violence has been provided. Neither in introduction section nor within methods a clear description of workplace physical violence can be found (in the terms of types of

incidents, perpetrators, victims, statistics). Moreover, Authors do not report any data for violence exposure in presented study besides the general prevalence in the previous 12 months (13.60%). It would be recommended to report the results for measures used (Workplace Violence Scale and Survey of Violence Experienced by Staff).

5. It would be also recommended to consider the exposure to physical violence as a correlate of PTSD (probably main correlate) or to control it in statistical analyses. It is known that trauma exposure is one of the strongest predictor of posttraumatic stress symptoms (APA, 2013). In the presented study the physical violence might be included into regression analysis along with already tested variables - it could be crucial to analyze the predicting effect of exposure to physical violence in tested model.

6. Authors inform that "A total of 2,706 valid questionnaires were returned, and the effective response rate was 84.25%. This study was only about physical violence, so the 368 healthcare workers are suitable for research" (p. 6). This information might be understood that only those workers, who experienced physical violence joined the study group. But at the same time an information: "the prevalence of physical violence in the previous 12 months was 13.60%" is included (p. 10). Thus it would be recommended to report clearly what was the inclusion criterium related to violence exposure.

7. Authors indicate that the convenience sampling method was applied while participants selection (p. 6). It would be important to clearly state that this is non-probability method and the findings shouldn't be generalized.

8. Authors indicate that 150 questionnaires were distributed and recovered during a pilot study. Are these data included into main study? (p. 6)

9. It would be appreciated if Authors provide all information on demographic variables (page 6, line 54).

10. In the description of PCL-C, Authors inform that "A total score of ≥ 50 is indicative of PTSD symptoms". It would be more precise to state that this score indicates the full PTSD diagnosis. The whole range of PCL-C scores define the intensity (low vs high) of PTSD symptoms, while the score above 50 defines clinical diagnosis of PTSD.

11. Each questionnaire description provide different range of information on subscales, dimensions, scoring, etc. A standardized description for all psychometric tools (the same manner for all questionnaires) would improve the clearness of information provided. Also, please include the references for the questionnaires. For TCSQ, the brief definition of positive and negative coping would be appreciated.

12. Authors inform that individuals who experienced other traumatic events than workplace-related violence were excluded from the study. It would be appreciated to include additional information if the indirect exposure to trauma was also considered. Healthcare workers are at high risk of indirect (secondary) traumatization (e.g. Cieslak, Anderson, et al., 2013; Meadors, et al., 2010). Indirect exposure to trauma refers to non-personal involvement into traumatic events through the confrontation with other people's

traumatic experiences, which may trigger the development of PTSD symptoms (APA, 2013). The issue of indirect traumatization could be addressed in the presented study.

13. P. 2, line 54: “Most of the participants (47.0%) did not appear to be have PTSD symptoms after experiencing physical violence.” – it would be suggested to address this confusing information, since the 47% is less than half, rather than “most”. Also please consider refining the expression “to be have PTSD symptoms”. Moreover Authors state that “The prevalence of PTSD symptoms among healthcare workers who experienced physical violence was high” (p. 3). It is suggested to address these contradictions.

14. P. 2, line 56: please consider refining “The healthcare workers adopted negative coping with physical violence was positively associated with the development of PTSD symptoms”. Also, are questions from TCSQ related to coping with violence at work or with coping style in general? The above sentence suggest that participants were asked, how do they cope with job-related violence particularly.

15. P. 3, line 49: “large sample size research could contribute to the generalization of our findings” – it would be important to notice that the possibility of generalization depends not particularly on the sample size but on its representatives.

16. In data analysis section Authors state that “Data were double entered after carefully checking and eliminating data that did not qualify for the statistical” (p. 9) – it seems not clear enough why some data did not qualify for the analysis.

17. P. 15: Authors indicate that “A previous study also reported that the healthcare workers of emergency department were the direct victims of workplace violence because they reported re-experiencing the violent event, followed by hyper-arousal, and avoidance. This finding might reflect the normal stress response of healthcare workers and support the notion that some healthcare workers might benefit from relaxation training and psychological interventions by professionals.” – it seems to be a bit unclear and confusing.

18. P. 17: “Healthcare workers who have experienced physical violence are more likely to develop PTSD symptoms” – it would be valuable to clearly indicate the reference group.

19. P. 17: Authors state that “These findings suggest that social support, coping styles, whether a person was exposed to physical violence, emotional instability, and anxious personality is closely related to PTSD symptoms.” However the statistics for the associations between exposure to violence and PTSD symptoms are not provided. Also it is unclear how to understand the term “closely related”.

20. While discussing the results on gender-related differences in the PTSD prevalence Authors state: “This might be attributed to gender differences in coping styles and social support.” (p. 16) - does it relate to this study findings? If so, it would be interesting to provide detailed data on these findings.

21. The discussion of results provided by Authors is somewhat

desultory. The discussion section seems to be mainly a broader summary of study results. Preferably it might be expanded by reference to theory of trauma and empirical evidences from other studies to explain how the results may be understood and why. Also, Authors conclude that the results may serve as a justification for implementing psychological interventions in workplace. However, the cross-sectional design of this study makes it hard to establish the causal relationship between tested variables and thus provide a recommendation for practice. For example, authors state that “hospitals could provide violence-related training for healthcare workers and provide psychological support or a “debriefing room” (p. 17), also: “as expected, social support was negatively associated with PTSD symptoms (...) and was a protective factor” (p. 3). It might be deduced that Authors suggest that enhancing support via workplace interventions may protect from development of PTSD in the aftermath of violence incidents. However in presented study the relationship between support and PTSD cannot be considered causally. Potentially the associations chain may alternatively look like that: violence exposure leads to the increase in PTSD and next to decrease in social support. Thus the conclusion on protective role of support based on the findings from this particular study might be misguided. The causal relationship and thus conclusions on possible interventions might be indeed assumed on the basis of some theoretical models (e.g. Conservation of Resources Theory, Hobfoll, 1989, or Social Cognitive Model of Trauma, Benight & Bandura, 2004), however Authors do not provide any relevant references.

22. The associations between social support and PTSD should be analyzed more in depth. At first it would be beneficial to analyze the subscales of SSRS separately, as they represent distinctive dimensions of support: subjective (perceived) support, objective (received) support, and utilization of social support. Secondly, the broader discussion of the results should be provided. According to Social Support Deterioration Model (Kaniasty, & Norris, 1995) and Social Support Deterioration Deterrence Model (Norris, & Kaniasty, 1996) the trajectories of support after traumatic events may differ depending on the type of support. It would be worthwhile to refer to theoretical models and previous findings while discussing support-PTSD associations. Also, Authors state: “The positive effects of social support on PTSD symptoms suggest that organizational and familial support has practical implications for interventions to promote psychological health” (p. 3 also mentioned at p. 16), however according to Authors description the SSRS scale does not distinguish different sources of social support. Also it is worth noticing that within SSRS description (p. 9) Authors indicate that the questionnaire “is a short measure of the social support individuals have received”, however only one SSRS subscale refer to received support.

23. Limitations section: although authors do enlist some important limitations, such as cross-sectional design of the study, the problem of low representativeness due to the convenience sampling thus limited generalizability of the results is not addressed (Authors state that: “results are specific to Chinese healthcare workers exposed to physical violence”, p. 17).

24. The check for the grammar, punctuation, misspellings, and language in general should be done, as parts of the text need some improvement (e.g.: p. 3 line 44, p. 6 line 34, sentence p. 2: “A total of 2706 participants from 39 public hospitals located in Heilongjiang,

Hebei, and Beijing Provinces of China (effective response rate = 84.25%).”, sentence p. 11: “The participants (21.2%) were considered to be at risk for later developing PTSD.”, sentence p. 12: “The criterion for PTSD that was the least frequently happening criterion for PTSD observed in the physical violence group was avoidance”, and other).

REFERENCE LIST

25. Th check whether all listed references correspond with the in-text citations is necessary (e.g. reference number 9, 13,14, 29 seems to be unrelated to the manuscript content).

TABLES

26. Table 1: as the SSRS scale for social support measures three distinct dimensions of support: subjective (perceived) support, objective (received) support, and utilization of social support/support-seeking behavior it would be relevant to report statistics for all three subscales.

27. Table 2: Term “PTSD” would be more precise than “PTSD symptoms” – the whole range of PCL-C scores define the intensity (low vs high) of PTSD symptoms, while the score above 50 defines clinical diagnosis of PTSD. Also term “civilian” is missing in the table footnote.

28. Table 3: It would be beneficial to include M and SD statistics for all listed variables and also the statistics for: (1) separate PTSD subscales, (2) separate SSRS subscales, and (3) physical violence exposure (Workplace Violence Scale and Survey of Violence Experienced by Staff), since Authors indicate the use of these measures in the Methods section of the manuscript.

29. Table 4: The expression “positive correlates of PTSD” in the table title seems to be confusing.

30. Table 5: missing β for negative coping in men.

31. Tables 3, 4, 5: it would be convenient to find explanations of acronyms below tables.

References:

American Psychiatric Association. (2000). Diagnostic and statistical manual of mental disorders (4th ed., text rev.). Washington, DC: American Psychiatric Association

American Psychiatric Association. (2013). Diagnostic and statistical manual of mental disorders (5th ed.). Washington, DC: American Psychiatric Association.

Benight, C. C., & Bandura, A. (2004). Social cognitive theory of posttraumatic recovery: The role of perceived self-efficacy. *Behaviour research and therapy*, 42(10), 1129-1148.

Cieslak, R., Anderson, V., Bock, J., Moore, B. A., Peterson, A. L., & Benight, C. C. (2013). Secondary traumatic stress among mental health providers working with the military: prevalence and its work– and exposure–related correlates. *The Journal of Nervous and*

	Mental Disease, 201(11), 917–925. http://doi.org/10.1097/NMD.0000000000000034 Hobfoll, S. E. (1989). Conservation of resources: A new attempt at conceptualizing stress. American psychologist, 44(3), 513. Kaniasty, K., & Norris, F. H. (1995). Mobilization and deterioration of social support following natural disasters. Current Directions in Psychological Science, 4(3), 94-98. Meadors, P., Lamson, A., Swanson, M., White, M., & Sira, N. (2010). Secondary traumatization in pediatric healthcare providers: Compassion fatigue, burnout, and secondary traumatic stress. OMEGA-Journal of Death and Dying, 60(2), 103-128. Norris, F. H., & Kaniasty, K. (1996). Received and perceived social support in times of stress: A test of the social support deterioration deterrence model. Journal of personality and social psychology, 71(3), 498.
--	---

VERSION 1 – AUTHOR RESPONSE

Reviewer: 1

Institution and Country: Azienda Ospedaliera di Perugia, Perugia (Italy)

Dear Dr Ramaciati,

Thank you very much for your valuable advice. We revised the manuscript according to your suggestion (The traces of change are represented in purple). Modify as follows :

1. Competing Interests: None declared
2. We revised the journal abbreviations in all references.

Hope you have a nice day.

Best wishes,

Lei Shi

Department of Health Management, School of Public Health, Harbin Medical University, China

Respond to the Reviewer's Comments :

Reviewer: 2

Institution and Country: SWPS University of Social Sciences and Humanities,
Department of Psychology

Dear Dr Lesnierowska,

Thank you very much for your valuable advice. We have learned a lot from these suggestions. Thank you again for the work that you have done for this manuscript. We revised the manuscript according to your suggestion (The traces of change are represented in blue). Modify as follows : Competing Interests: None declared

1. Advice: Please note, that the most current DSM edition is DSM-5 (APA, 2013), which should be also discussed in the context of trauma symptoms described in this manuscript.

We changed to "Post-traumatic stress disorder (PTSD) is a psychological state of imbalance, characterized by a series of chronic emotional reactions to a traumatic event, including re-experiencing, avoidance, and heightened arousal, as outlined in the Diagnostic and Statistical Manual

of Psychiatric Disorders-4th edition (DSM-IV).¹⁻³ However, the criteria for PTSD in the manual's fifth edition (DSM-5) include not three but four symptom clusters: including re-experiencing, avoidance, negative alterations in mood and cognition, and hyperarousal. It is worth noting that PTSD has shifted from its classification as an anxiety disorder in the DSM-IV to a new category of Trauma and Stress-related Disorders in the DSM-5."

2. Advice: It is recommended to expand the conceptual part in terms of why and how posttraumatic stress symptoms develop, especially among this particular occupational group. Provision of meta-analysis data would be highly appreciated.

We changed to " PTSD symptoms and the full range of criteria comprising a PTSD diagnosis have been observed in rescue and ambulance personnel. Healthcare workers typically are exposed to two types of trauma in the hospital setting: direct (personal involvement in traumatic events through confrontations resulting in their own traumatic experiences e.g., workplace violence) and indirect (non-personal involvement in traumatic events through others' confrontations resulting in other people's traumatic experiences e.g., witnessing other people's direct experiences of workplace violence, caring for dying patients, and threats of severe injury or exposure to trauma). In the present study, a traumatic event refers to a healthcare worker's exposure to physical violence in the workplace. Workplace violence is divided into physical and psychological violence. Physical violence causes more serious physical and psychological damage (e.g., PTSD, anxiety, fear, and depression) than other forms of violence. " "Demographic variables (e.g., age, gender, and educational level) and psychological and social variables (e.g., personality, coping style, and social support) have been found to be significantly associated with cancer-related PTSD symptoms. Previous studies have found that the risk of PTSD was most strongly associated with neuroticism and problem-focused coping strategies in the general population. Neuroticism was the most critical personality dimension in predicting PTSD, and avoidant coping and social support mediated the relationship between neuroticism and PTSD .in a high proportion of adult burn survivors.³⁶ Social support has been reported to play a significant role in helping nurses cope with work-related stress. A meta-analysis indicated that work-related critical incidents were positively related to PTSD symptoms"

3. Advice: Authors do not provide a reference and data on prevalence of PTSD symptoms among healthcare workers. It would be beneficial to refer to relevant results (preferably meta-analytical). We changed to "Several studies have estimated the prevalence of PTSD among emergency department staff to range from 10% to 25%. Noelle Robertson and Andrew Perry conducted a systematic review of PTSD research investigations; the results showed that the prevalence of PTSD ranged from 8% to 29% among different hospital-based departments."

4. Advice: Neither in introduction section nor within methods a clear description of workplace physical violence can be found (in the terms of types of incidents, perpetrators, victims, statistics). It would be recommended to report the results for measures used.

We changed to Introduction: "Physical violence refers to the use of physical force against an individual or a group, and can lead to physical, psychological, or sexual harm; it includes hitting, shooting, kicking, slapping, pushing, biting, pinching, wounding using sharp objects, and sexual assault and rape. Approximately 50% of healthcare workers have experienced at least one violent incident during their working lives. During the past 12 months, the incidence rate of physical violence for nurses in different countries has ranged from 9.1% to 56.0%. The results of a systematic review of studies conducted in Iran indicated that the most common types of physical violence experienced by 43% of participants were pushing or pinching. In China, physician-patient conflicts present a growing trend, with an increase in the number of healthcare workers killed by patients or their relatives to 24, and an increase in injures from 2003 to 2013. " Results: "During the past 12 months, the prevalence of physical violence and psychological violence toward healthcare workers were 13.60% (368/2706) and 59.64% (1614/2706), respectively. The respondents reported that the patients' relatives were the main perpetrators (67.4%, n = 248), followed by the patients (23.6%, n = 87). "

5. Advice: It would be also recommended to consider the exposure to physical violence as a correlate of PTSD (probably main correlate) or to control it in statistical analyses. In the presented study the physical violence might be included into regression analysis along with already tested variables - it could be crucial to analyze the predicting effect of exposure to physical violence in tested model. We changed to " Table 4 and Table 5 showed that physical violence are analyzed. " Table 4: Physical violence was positively associated with PTSD symptoms ($r = 0.259$, $P < 0.001$). Table 5: As shown in Block 2, physical violence was positively associated with PTSD symptoms ($\beta = 1.216$, $P < 0.001$). As shown in Block 3, positive coping as measured by the TCSQ was negatively associated with PTSD symptoms ($\beta = -0.327$, $P = 0.002$), whereas, negative coping was positively associated with PTSD symptoms in the regression model ($\beta = 0.353$, $P = 0.001$). Furthermore, gender had a significant effect on PTSD symptoms, and men were more vulnerable to PTSD symptoms than women (Table 1). Therefore, we explored the potential correlates of PTSD symptoms in men and women (Table 6). As shown in Block 3, among the women, positive coping as measured by the TCSQ was significantly associated with PTSD symptoms ($\beta = -0.376$, $P = 0.001$), but the effect of positive coping was not significant in men.

6. Advice: Authors inform that "A total of 2,706 valid questionnaires were returned, and the effective response rate was 84.25%. This study was only about physical violence, so the 368 healthcare workers are suitable for research". This information might be understood that only those workers, who experienced physical violence joined the study group. Thus it would be recommended to report clearly what was the inclusion criterium related to violence exposure.

We changed to " This study's focus was only on PTSD symptoms among healthcare workers exposed to physical violence; thus, only 368 responses were considered valid data and were analyzed in the present study."

"The inclusion criteria for participation in this study were as follows: (1) at least one year of work experience; (2) voluntary participation; (3) participation would not affect the participation's work; and (4) experience of physical violence in the previous 12 months. Individuals were excluded if they (1) had received any psychological treatment after experiencing physical violence; (2) experienced other traumatic events, including workplace psychological violence or serious life events (e.g., domestic violence or attacks by criminals), serious accidents (e.g., fires, explosions, or traffic accidents), natural disasters (e.g., typhoons, earthquakes, or floods), or (3) were indirectly exposed to trauma, (e.g., witnessing other people experience traumatic events)."

7. Advice: Authors indicate that the convenience sampling method was applied while participants selection (p. 6). It would be important to clearly state that this is non-probability method and the findings shouldn't be generalized.

We changed to "The 39 public hospitals that served as the research settings were chosen using convenience sampling method (convenience sampling method is a non-probability method, and the findings should not be generalized). "

8. Advice: Authors indicate that 150 questionnaires were distributed and recovered during a pilot study. Are these data included into main study?

We changed to "A total of 150 questionnaires were distributed and returned (these data were excluded from the main study)."

9. Advice: It would be appreciated if Authors provide all information on demographic variables (page 6, line 54).

We changed to "Demographic data on the healthcare workers were collected, including gender, age, marital status, educational status, professional title, department, occupation, and work experience. Age was categorized as ≤ 30 , 31-50, and ≥ 51 years old. Marital status was categorized as married and single/ divorced/widowed. Educational status was classified as junior college or below, undergraduate, and graduate. Occupation was divided into three groups: physician, nurse, and

medical technician. Professional title was categorized as primary, intermediate, and senior. Department was classified as emergency department, internal medicine, surgery, obstetrics and gynecology, pediatrics, and other. Work experience was divided into four categories: ≤ 4 , 5–10, 11–20, and ≥ 21 years.”

10. Advice: In the description of PCL-C, Authors inform that “A total score of ≥ 50 is indicative of PTSD symptoms”. It would be more precise to state that this score indicates the full PTSD diagnosis. The whole range of PCL-C scores define the intensity (low vs high) of PTSD symptoms, while the score above 50 defines clinical diagnosis of PTSD.

We changed to “A total score ≥ 50 is indicative of the full PTSD diagnosis (sensitivity = 0.82; specificity = 0.83; kappa = 0.64).” “Table 3: Criteria met for the full PTSD diagnosis ”

11. Advice: Each questionnaire description provide different range of information on subscales, dimensions, scoring, etc. A standardized description for all psychometric tools (the same manner for all questionnaires) would improve the clearness of information provided. Also, please include the references for the questionnaires. For TCSQ, the brief definition of positive and negative coping would be appreciated.

We perfected each questionnaire description according to your suggestion.“ Positive coping refers to individuals who, when faced with a problem, tend to deal with it in a positive way, and are able to quickly forget unpleasant aspects. Negative coping refers to the tendency to use negative coping methods to deal with problems and vent frustrations to other people, which makes it is easier to ignore unpleasant thoughts. For example, when conflicts with others, arise, individuals who use negative coping will ignore the opposing side for a long time.”

12. Advice: It would be appreciated to include additional information if the indirect exposure to trauma was also considered. Healthcare workers are at high risk of indirect (secondary) traumatization (e.g. Cieslak, Anderson, et al., 2013; Meadors, et al., 2010). Indirect exposure to trauma refers to non–personal involvement into traumatic events through the confrontation with other people's traumatic experiences, which may trigger the development of PTSD symptoms (APA, 2013). The issue of indirect traumatization could be addressed in the presented study.

We changed to “Healthcare workers typically are exposed to two types of trauma in the hospital setting: direct (personal involvement in traumatic events through confrontations resulting in their own traumatic experiences e.g., workplace violence) and indirect (non-personal involvement in traumatic events through others' confrontations resulting in other people's traumatic experiences e.g., witnessing other people's direct experiences of workplace violence, caring for dying patients, and threats of severe injury or exposure to trauma) .”

13. Advice: P. 2, line 54: “Most of the participants (47.0%) did not appear to be have PTSD symptoms after experiencing physical violence.” – it would be suggested to address this confusing information, since the 47% is less than half, rather than “most”. Also please consider refining the expression “to be have PTSD symptoms”. Moreover Authors state that “The prevalence of PTSD symptoms among healthcare workers who experienced physical violence was high” (p. 3). It is suggested to address these contradictions.

We changed to “ Most of the victims of physical violence (50.80%) did not exhibit PTSD symptoms based on their PCL-C scores, and 47.0% did not manifest the diagnostic criteria for PTSD after experiencing physical violence. ” “The prevalence of PTSD among the victims was similar to that found in Atlanta.”

14. Advice: P. 2, line 56: please consider refining “The healthcare workers adopted negative coping with physical violence was positively associated with the development of PTSD symptoms”.

We changed to “ The healthcare workers' coping styles influenced the development of PTSD symptoms. ” “The questions are from TCSQ related to coping style in general.”

15. Advice: It would be important to notice that the possibility of generalization depends not particularly on the sample size but on its representatives.
We changed to “Our study was conducted at 39 public hospitals in three provinces using convenience sampling. Therefore, the representativeness of the sample is limited.”

16. Advice: In data analysis section Authors state that “Data were double entered after carefully checking and eliminating data that did not qualify for the statistical” (p. 9) – it seems not clear enough why some data did not qualify for the analysis.
We changed to “We eliminated the questions with missing data or quality issues. To ensure accuracy, two trained personnel entered the data after all the surveys were completed.”

17. Advice: P. 15: Authors indicate that “A previous study also reported that the healthcare workers of emergency department were the direct victims of workplace violence because they reported re-experiencing the violent event, followed by hyper-arousal, and avoidance. This finding might reflect the normal stress response of healthcare workers and support the notion that some healthcare workers might benefit from relaxation training and psychological interventions by professionals.” – it seems to be a bit unclear and confusing.
We changed to “ The most commonly observed PTSD symptoms was re-experiencing (45.1%), followed by hyper-arousal (37.8%), and then avoidance (35.1%). A previous study also reported that healthcare workers in an emergency department were victims of direct workplace violence because they reported re-experiencing the violent event, followed by hyper-arousal and avoidance.”

18. Advice: P. 17: “Healthcare workers who have experienced physical violence are more likely to develop PTSD symptoms” – it would be valuable to clearly indicate the reference group.
We changed to “The prevalence of PTSD among healthcare workers exposed physical violence was similar to that in Atlanta. ”

19. Advice: P. 17: Authors state that “These findings suggest that social support, coping styles, whether a person was exposed to physical violence, emotional instability, and anxious personality is closely related to PTSD symptoms.” However the statistics for the associations between exposure to violence and PTSD symptoms are not provided. Also it is unclear how to understand the term “closely related”.
We added physical violence data to Table 4 and Table 5. We changed to “These findings suggest that social support, exposure to physical violence, and coping styles are closely related to PTSD symptoms. ”

20. Advice: While discussing the results on gender-related differences in the PTSD prevalence Authors state: “This might be attributed to gender differences in coping styles and social support.” (p. 16) - does it relate to this study findings? If so, it would be interesting to provide detailed data on these findings.
It does not relate to this study findings. We changed to “ This finding might be attributed to gender differences in responses to different traumatic events and in social networks.61-62”

21. Advice: The discussion of results provided by Authors is somewhat desultory. Also, Authors conclude that the results may serve as a justification for implementing psychological interventions in workplace. However, the cross-sectional design of this study makes it hard to establish the causal relationship between tested variables and thus provide a recommendation for practice. For example, authors state that “hospitals could provide violence-related training for healthcare workers and provide psychological support or a “debriefing room” (p. 17), also: “as expected, social support was negatively associated with PTSD symptoms (..) and was a protective factor” (p. 3). It might be deduced that Authors suggest that enhancing support via workplace interventions may protect from development of

PTSD in the aftermath of violence incidents. However in presented study the relationship between support and PTSD cannot be considered causally. Potentially the associations chain may alternatively look like that: violence exposure leads to the increase in PTSD and next to decrease in social support. Thus the conclusion on protective role of support based on the findings from this particular study might be misguided. The causal relationship and thus conclusions on possible interventions might be indeed assumed on the basis of some theoretical models (e.g. Conservation of Resources Theory, Hobfoll, 1989, or Social Cognitive Model of Trauma, Benight & Bandura, 2004), however Authors do not provide any relevant references.

We re-written the discussion section according to your suggestions. The interventions might be indeed assumed on the basis of some theoretical models. We changed to “Therefore, we recommend interventions based on the social cognitive theory.⁶³ For example, social support has been found to be an important protective factor in reducing stress and depression, and improving health.⁶³ After the occurrence of a traumatic event, enabling function of social support can enhance self-efficacy, thereby promoting recovery from the trauma.⁶³”

22. Advice: The associations between social support and PTSD should be analyzed more in depth. At first it would be beneficial to analyze the subscales of SSRS separately, as they represent distinctive dimensions of support: subjective (perceived) support, objective (received) support, and utilization of social support. Secondly, the broader discussion of the results should be provided. According to Social Support Deterioration Model (Kaniasty, & Norris, 1995) and Social Support Deterioration Deterrence Model (Norris, & Kaniasty, 1996) the trajectories of support after traumatic events may differ depending on the type of support. It would be worthwhile to refer to theoretical models and previous findings while discussing support-PTSD associations.

We changed to “As shown in the results of the Pearson’s correlations and the hierarchical regression analysis, social support had a significant negative association with PTSD symptoms, and this finding is consistent with other research.^{9 36 52 53} The level of PTSD symptoms was significantly and negatively correlated with the healthcare workers’ scores for objective support and utilization of support. A previous study found that the Deterioration Model of Social Support has been useful in discriminating the potential of stressors to reduce support.⁵⁸ They found that disaster-induced erosion of perceived social support increased symptoms of depression among both primary and secondary victims; the loss of perceived social support also mediated psychological consequences.⁵⁹ The Deterioration Deterrence Model of Social Support which is similar to support-mobilization models, has been used to explain how the perceived deterioration of social support can be counteracted by higher levels of received social support.⁵⁹⁻⁶⁰ If post-disaster support mobilization is implemented, stress should be positively correlated with received support. At the same time, received support should be positively related to perceived support. Therefore, the receipt of support should suppress a negative relationship between stress and perceived support.⁵⁹⁻⁶⁰ Victims of physical violence should be encouraged not to abandon their daily social activities because these activities have many important functions (e.g., they help people understand the needs of network members and inspire their participation in helping).⁶⁰ Daily contact is the most natural forum for sharing experiences, which might suppress negative emotions, provide opportunities for social comparison, and maintain a sense of friendship and feelings of being accepted.⁶⁰ It is important to recognize that stress caused by violence is persistent. Yet, a supportive hospital environment can help individuals cope with a wide range of stressful events and serve as a buffer against their negative health effects.⁵⁹⁻⁶⁰”

Advice: Authors indicate that the questionnaire “is a short measure of the social support individuals have received”, however only one SSRS subscale refer to received support.

We changed to “Subjective support refers to an individual’s emotional experience of being respected, supported, and understood by their social group, and it is closely related to the individual’s subjective feelings. Objective support refers to visible support, including material and direct assistance, social networks, group relationships, and the individual’s degree of participation in societal activities with family, friends, and colleagues (e.g., marriage).”

23. Advice: Limitations section: although authors do enlist some important limitations, such as cross-sectional design of the study, the problem of low representativeness due to the convenience sampling thus limited generalizability of the results is not addressed.

We changed to “Finally, our results are specific to Chinese healthcare workers exposed to physical violence in the past 12 months; thus, the low representativeness of the sample due to the use of convenience sampling limits the generalizability of the results. The inclusion of healthcare workers from a wider range of careers in a more representative sample should contribute to the ability to generalize the results of future studies.”

24. Advice: The check for the grammar, punctuation, misspellings, and language in general should be done, as parts of the text need some improvement.

p. 3 line 44, p. 6 line 34, sentence p. 2: “A total of 2706 participants from 39 public hospitals located in Heilongjiang, Hebei, and Beijing Provinces of China (effective response rate = 84.25%).” is changed to “A total of 2,706 valid questionnaires were returned, and the effective response rate was 84.25%.” sentence p. 11: “The participants (21.2%) were considered to be at risk for later developing PTSD.” is changed to “According to their scores on the PCL-C, 103 victims (28.0%) met the full criteria for a PTSD diagnosis and 21.2% of victims were at risk for developing PTSD.”

sentence p. 12: “The criterion for PTSD that was the least frequently happening criterion for PTSD observed in the physical violence group was avoidance” is changed to “Re-experiencing was the most frequently observed criterion for PTSD observed among the victims (45.1%), followed by hyperarousal (37.8%).”

Other changes were seen in red.

25. Advice: The check whether all listed references correspond with the in-text citations is necessary (e.g. reference number 9, 13, 14, 29 seems to be unrelated to the manuscript content).

We deleted the references that were not relevant to the manuscript (e.g., reference number 9, 13, 14, 29).

26. Advice: Table 1: as the SSRS scale for social support measures three distinct dimensions of support: subjective (perceived) support, objective (received) support, and utilization of social support/support-seeking behavior it would be relevant to report statistics for all three subscales.

We added report statistics for all three subscales of SSRS in Table 1.

27. Advice: Table 2: Term “PTSD” would be more precise than “PTSD symptoms” – the whole range of PCL-C scores define the intensity (low vs high) of PTSD symptoms, while the score above 50 defines clinical diagnosis of PTSD. Also term “civilian” is missing in the table footnote.

We changed to “Criteria met for the full PTSD diagnosis (50-85)” We revised PCL-C in the table footnote.

28. Advice: Table 3: It would be beneficial to include M and SD statistics for all listed variables and also the statistics for: (1) separate PTSD subscales, (2) separate SSRS subscales, and (3) physical violence exposure

We added the M and SD statistics for all listed variables and also the statistics for: (1) separate PTSD subscales, (2) separate SSRS subscales, and (3) physical violence exposure.

29. Advice: Table 4: The expression “positive correlates of PTSD” in the table title seems to be confusing.

We changed to “Hierarchical regression for exploring the correlates of PTSD symptoms.”

30. Advice: Table 5: missing for negative coping in men.

We added the negative coping in men in Table 6.

31. Advice: Tables 3, 4, 5: it would be convenient to find explanations of acronyms below tables. We added explanations of acronyms below tables (Tables 2, 3, 4, 5,6).

References

4. Association AP. Diagnostic and statistical manual of mental disorders fifth edition, text revision (DSM-V). 2013.
12. Alexander DA, Klein S. Ambulance personnel and critical incidents: impact of accident and emergency work on mental health and emotional well-being. *Brit J Psychia*2001;178(1):76.
13. Jonsson A, Segesten K, Mattsson B. Post-traumatic stress among Swedish ambulance personnel. *Emerg Med J* 2003;20(1):79-84.
14. Laposa JM, Alden LE, Fullerton LM. Work stress and posttraumatic stress disorder in ED nurses /personnel. *J Emerg Nurs* 2003;29(1):23-28.
15. Shoji K, Lesnierowska M, Smoktunowicz E, et al. What Comes First, Job Burnout or Secondary Traumatic Stress? Findings from Two Longitudinal Studies from the U.S. and Poland. *Plos One* 2015;10(8):e0136730.
16. Boer JD, Lok A, Verlaat EV, et al. Work-related critical incidents in hospital-based health care providers and the risk of post-traumatic stress symptoms, anxiety, and depression: A meta-analysis. *Soc Sci Med* 2011;73(2):316326.
17. Krug EG, Mercy JA, Dahlberg LL, et al. The world report on violence and health. *Lancet* 2002;360(9339):1083.
18. Ray MM. The dark side of the job: Violence in the emergency department. *J Emerg Nurs* 2007;33 :257-261.
21. Harrell, Erika. Workplace Violence, 1993-2009: National Crime Victimization Survey and the Census of Fatal Occupational Injuries. *Digitalcommons* 2011.
22. Magnavita N, Heponiemi T. Workplace violence against nursing students and nurses: an Italian experience. *J Nurs Scholarsh* 2011;43(2):203-210.
23. Fute M, Mengesha ZB, Wakgari N, et al. High prevalence of workplace violence among nurses working at public health facilities in Southern Ethiopia. *BMC Nurs* 2015;14:9.
24. Al-Omari H. Physical and verbal workplace violence against nurses in Jordan. *Int Nurs Rev* 2015;62(1):111–118.
25. Schablon A, Zeh A, Wendeler D, et al. Frequency and consequences of violence and aggression towards employees in the German healthcare and welfare system: a cross-sectional study. *BMJ Open* 2011;2(5):421-421.
26. Najafi F, Fallahi-Khoshknab M, Dalvandi A, et al. Workplace violence against Iranian nurses: A systematic review. *J Health Promot Manage* 2014;3:72-85.
27. Pan Y, Yang XH, He JP, et al. To be or not to be a doctor, that is the question: a review of serious incidents of violence against doctors in China from 2003–2013. *J public health* 2015;23(2):111-116
31. Robertson N, Perry A. Institutionally based health care workers' exposure to traumatogenic events: Systematic review of PTSD presentation. *Journal of Traumatic Stress* 2010;23(3):417- 420.
34. Perrin M, Vandeleur C L, Castelao E, et al. Determinants of the development of post-traumatic stress disorder, in the general population. *Soc Psych Psych Epid* 2014;49(3):447-457.
35. Breslau N, Davis G C, Andreski P, et al. Traumatic events and posttraumatic stress disorder in an

- urban population of young adults. *Arch Gen Psychiatry* 1991;48(3):216-222.
36. Lawrence JW, Fauerbach JA. Personality, coping, chronic stress, social support and PTSD symptoms among adult burn survivors: a path analysis. *J Burn Care Rehabil* 2003;24(1):63.
37. Kerasiotis B, Motta RW. Assessment of PTSD symptoms in emergency room, intensive care unit, and general floor nurses. *Int J Emerg Ment Health* 2004;6(3):121-133.
38. Boer JD, Lok A, Verlaet EV, et al. Work-related critical incidents in hospital-based health care providers and the risk of post-traumatic stress symptoms, anxiety, and depression: A meta-analysis. *Soc Sci Med* 2011;73(2):316-326.
39. Shoji K, Lesnierowska M, Smoktunowicz E, et al. What Comes First, Job Burnout or Secondary Traumatic Stress? Findings from Two Longitudinal Studies from the U.S. and Poland. *Plos One* 2015;10(8):e0136730.
40. Cieslak R, Anderson V, Bock J, et al. Secondary traumatic stress among mental health providers working with the military: Prevalence and its work- and exposure-related correlates. *J Nerv Ment Dis* 2013;201(11):917-25.
43. Battles ED. An exploration of post-traumatic stress disorder in emergency nurses following Hurricane Katrina. *J Emerg Nurs* 2007;33(4):314-318.
44. Dirk Richter K B. Post-traumatic stress disorder following patient assaults among staff members of mental health hospitals: a prospective longitudinal study. *BMC Psychiatry* 2006;6(1):15.
45. Weathers FW, Litz BT, Herman DS, et al. The PTSD checklist: reliability, validity & diagnostic utility. Paper presented at the Annual Meeting of the International Society for Traumatic Stress Studies, San Antonio, TX 1993.
46. Wu Z, Xu J, Sui Y. Posttraumatic stress disorder and posttraumatic growth coexistence and the risk factors in Wenchuan earthquake survivors. *Psychiatry Res* 2016; 237:49-54.
50. Zhang L, Zhang J X, Wen-Juan X U, et al. An evaluation of validity and reliability on the trait coping style questionnaire(TCSQ) using for Chinese soldier. *Chinese Journal of Disease Control & Prevention* 2010.
51. Li-Na L I, Zhao Y, Cui X J, et al. The Influence Factors and Status on Male Vicarious Trauma of Disaster. *Chinese General Practice* 2014.
55. Mealer ML, Shelton A, Berg B, et al. Increased prevalence of post-traumatic stress disorder symptoms in critical care nurses. *Am J Resp Crit Care* 2007;175(175):693-697.
59. Kaniasty K, Norris FH. Mobilization and deterioration of social support following natural disasters. *Current Directions in Psychological Science* 1995;4(3):94-98.
60. Norris FH, Kaniasty K. Received and perceived social support in times of stress: a test of the social support deterioration deterrence model. *Journal of Personality & Social Psychology* 1996;71(3):498-511.
61. Freedman SA, Gluck N, Tuval-Mashiach R, et al. Gender differences in responses to traumatic events: a prospective study. *J Trauma Stress* 2002;15:407-413.
62. Pratchett LC, Pelcovitz MR, Yehuda R. Trauma and violence: are women the weaker sex? *Psychiatr Clin North Am* 2010; 33: 465-474.
63. Benight CC, Bandura A. Social cognitive theory of posttraumatic recovery: the role of perceived

VERSION 2 – REVIEW

REVIEWER	Magdalena Lesnierowska SWPS University of Social Sciences and Humanities, Poland
REVIEW RETURNED	05-Jun-2017

GENERAL COMMENTS	The revised version of the manuscript entitled "Prevalence and correlates of symptoms of post-traumatic stress disorder among Chinese healthcare workers exposed to physical violence: a cross-sectional study" presents an excellent development especially in terms of conceptualization, literature review, variables operationalization, discussion. Authors addressed comments or suggestions aptly and comprehensively. Few final remarks: 1. The sentence in abstract (p. 3): 'The prevalence of PTSD among the victims was similar to that found in Atlant' seems to be a bit out of context and it is hard to understand the reference. It could be beneficial to consider rephrasing or omitting it.2. Possible limitations of representativeness due to applying convenience sampling method could be discussed in limitation section only (instead of method section, p. 6).3. It would be suggested not to enumerate items for PCL, TCSQ, SSRS (p. 9- 11).
--

VERSION 2 – AUTHOR RESPONSE

Respond to the Reviewer's Comments :

Reviewer: 2

Institution and Country: SWPS University of Social Sciences and Humanities,
Department of Psychology

Dear Dr Lesnierowska,

Thank you very much for your valuable advice. Thank you again for the work that you have done for this manuscript. We revised the manuscript according to your suggestion (The traces of change are represented in blue). Modify as follows :

1. Competing interests None declared.(p. 23)

2. Advice: The sentence in abstract (p. 3): 'The prevalence of PTSD among the victims was similar to that found in Atlant' seems to be a bit out of context and it is hard to understand the reference. It could be beneficial to consider rephrasing or omitting it.

We changed to "The results suggest that the aftermath of physical violence contributes to current prevalence of PTSD. "

3. Advice: Possible limitations of representativeness due to applying convenience sampling method could be discussed in limitation section only (instead of method section, p. 6).

We removed the limitations of convenient sampling in the method section and discussed in limitation section only. We changed to "Finally, convenience sampling is a non-probability sampling method and the results of this study are specific to Chinese healthcare workers exposed to physical violence in the past 12-months.Thus, the low representativeness of the sample due to the use convenience sampling limits the generalizability of the results. "

4. Advice: It would be suggested not to enumerate items for PCL, TCSQ, SSRS (p. 9- 11).
We deleted enumerate items for PCL, TCSQ, SSRS.